# Adenomyosis and Abnormal Uterine Bleeding: Review of the Evidence

**DOI:** 10.3390/biom14060616

**Published:** 2024-05-23

**Authors:** Marwan Habiba, Sun-Wei Guo, Giuseppe Benagiano

**Affiliations:** 1Department of Health Sciences, University of Leicester, Leicester LE1 7RH, UK; 2Women and Perinatal Services, Leicester Royal Infirmary, Leicester LE1 5WW, UK; 3Research Institute, Shanghai Obstetrics and Gynecology Hospital, Fudan University, Shanghai 200011, China; hoxa10@outlook.com; 4Faculty of Medicine and Surgery, “Sapienza” University of Rome, 00161 Rome, Italy; giuseppe.benagiano@uniroma1.it; 5Geneva Foundation for Medical Education and Research, 1202 Geneva, Switzerland

**Keywords:** adenomyosis, abnormal uterine bleeding, menorrhagia, uterine disease

## Abstract

Background: Thomas Cullen described bleeding abnormalities and dysmenorrhea as the “expected” presentations of adenomyomas. Adenomyosis is included within the FIGO classification of structural causes of abnormal uterine bleeding (AUB). Nevertheless, this long-standing association has been questioned by some authors who reported a high incidence of adenomyosis in uteri removed for indications other than AUB or dysmenorrhea. Here, we examine evidence for the link between adenomyosis and AUB. Methods: A comprehensive Medline literature review of all publications to October 2023. Results: Sixty-three articles were identified and included in the review. Despite a large body of studies, the available literature does not provide conclusive evidence of a link between adenomyosis and AUB. This is because of unsuitable study design, or poor characterization of the study population or of the inclusion or exclusion criteria. Additional challenges arise because of the lack of agreed criteria for diagnosing adenomyosis and the often absence of detailed assessment of menstrual blood loss. Adenomyosis often coexists with other conditions that have also been linked to similar symptoms, and many cases of adenomyosis are asymptomatic. Conclusion: Most of the existing literature and studies that addressed treatment outcome of adenomyosis started from the premise that a link between the condition and AUB had been proven. Yet, published information shows that aspects such a relationship is still uncertain. Further research is needed to address the relation between AUB and adenomyosis burden (or subtypes), distribution, and concomitant pathology.

## 1. Introduction

Adenomyosis is defined as the presence of endometrial tissue within the myometrium [1]. Whilst there are no pathognomonic symptoms of the condition, it has been associated with heavy menstrual bleeding (HMB) and dysmenorrhea [2,3]. The link in the medical literature between bleeding abnormalities and adenomyosis dates to early investigators, such Thomas Cullen who, in 1908, simply described bleeding abnormalities and dysmenorrhea as the “expected” presentations of adenomyomas [4]. Almost a century later, despite acknowledging that the relation between adenomyosis and the genesis of abnormal uterine bleeding (AUB) is unclear, the International Federation of Obstetrics and Gynecology (FIGO) PALM-COEIN framework included adenomyosis as a structural cause of AUB [5]. This classification was maintained in FIGO’s second iteration [6]. Attributing symptoms is challenging because of the lack of agreement on the exact definition and classification of adenomyosis [7] and because gynecological symptoms linked to adenomyosis are common and can occur in the presence, as well as in the absence, of demonstrable lesions [8]. Adenomyosis also commonly coexists with other uterine diseases such as fibroids, which renders symptom attribution difficult. One large study reported concomitant pathology, mostly fibroids, endometrial hyperplasia, and carcinoma or endometriosis in 60.2% of cases [9].

A large percentage of uteri removed at hysterectomies performed because of AUB are reported to have adenomyosis either as the sole or the main pathology, or in association with other lesions. It is often pointed out that adenomyosis is under-reported on routine histological examination, but there remains disagreement about cut-off points for histological diagnosis [9]. Thus, despite increased use of noninvasive imaging, the lack of agreement on the gold standard for diagnosis or classification will necessarily continue to pose a challenge. Published research takes limited, if any, account of disease distribution or lesion density.

The purpose of this review is to examine the published literature that references the relation between adenomyosis and AUB and to assess whether the introduction of the FIGO classification has enabled better understanding of this relation more than a decade since it was first proposed.

## 2. Material and Methods

We undertook a PubMed literature search on 4 October 2023 using the following search terms: uterine bleeding, (or) uterine hemorrhage, (or) menstrual disorder, (or) menstrual disturbances, (or) menorrhagia, (or) intermenstrual bleeding, (or) duration of bleeding as either text or indexed words. These yielded a total of 27,531 articles. These were then linked to articles identified using the term adenomyosis as a search item (n = 3801). Combining the two groups yielded 705 articles which were searched manually using the abstract and title. Articles not relevant to the question were eliminated, leaving 95 publications which were retrieved in full text. Additional references were identified from the list of references of the retrieved articles. Relevant articles (n = 63) were reviewed. Thirty-three of the identified references were published in the years following the publication of the FIGO classification, with 21 of these related to treatment outcomes.

## 3. Results

The search identified 28 comparative studies or case series which included 18,153 patients, with or without adenomyosis (Table 1). There were 10 publications that addressed the response to hormone preparations (Table 2), 8 studies that addressed the response to the levonorgestrel intrauterine system (LNG-IUS) (Table 3), and 17 studies that addressed response to surgery (Table 4).

Twelve studies in Table 1 were published after 2012, but only two of these make reference to the FIGO classification.

### 3.1. Comparative Studies and Case Series (Table 1)

There is considerable overlap, but the studies included here may be broadly grouped into those that included estimates of prevalence of adenomyosis in groups of women with common features or studies that attempted to link symptoms to the distribution of adenomyosis and a group. There were also studies that did not fit under either of these categories. Benson et al. [10] identified 701 (21.4%) cases of adenomyosis (sole pathology in 112 cases) among 3276 hysterectomies from premenopausal women aged <50 years. Adenomyosis was discovered incidentally, such as in cases of prolapse in 245 uteri. Fibroids were present in 56.6% of cases with adenomyosis. The most frequent menstrual complaint in this group was menorrhagia, which was the sole symptom in 43/112 (38.4%) cases where adenomyosis was the sole pathology and in 98/344 (28.5%) patients with associated pathology, whilst 34.9% of cases of adenomyosis were asymptomatic. Menometrorrhagia was less common, and metrorrhagia was rare. Thus, no definite conclusion could be reached of the relation between adenomyosis and AUB.

Molitor [11] reported adenomyosis in 281/3207 (8.8%) hysterectomy specimens, of which 81 (28.8%) were from asymptomatic women. In the asymptomatic group, adenomyosis was confined to the inner third of myometrium in 38, extended to the inner two-thirds in 33, and involved the whole myometrium in 10 cases. Fibroids coexisted with adenomyosis in 108 (38.5%) cases. The most common symptom was menorrhagia, followed by metrorrhagia, then by pain and dysmenorrhea alone or, less frequently, in combination. The indications for hysterectomy and the breakdown of symptoms in women with adenomyosis alone are not provided. The incidence of adenomyosis in this study is lower than other studies and there was a high incidence of asymptomatic cases.

Vercellini et al. identified adenomyosis in 24.9% of 1334 hysterectomy specimens [12]. The study, which relied on routine histological assessment of removed samples, did not support a link between adenomyosis and menorrhagia as there was no statistically significant difference in the incidence of adenomyosis between women with fibroids and menorrhagia (23.3%), prolapse (25.7%), ovarian cysts (21.4%), cervical cancer (19%), endometrial cancer (28.2%), ovarian cancer (28.1%), and other indications for hysterectomy (24.7%). It is not possible to disentangle the impact of adenomyosis and fibroids, as both were grouped together in the analysis. In common with many published studies, no clear rationale was provided for the chosen cut-off point for the diagnosis, and the study did not distinguish women based on menopausal status. Thus, the term “menorrhagia” seems to have been used to refer to any abnormalities of bleeding leading to hysterectomy. Furthermore, similar to many other studies, there was no control for variations in the approach to tissue sampling, the diagnostic cut-off points between pathologists, or in relation to cases with or without malignancy. The study has significant limitations, but the incidence of adenomyosis was similar, irrespective of the indication for hysterectomy.

Vavilis et al. [13] set out to estimate the frequency and risk factors for adenomyosis using the clinical records of 594 women undergoing hysterectomy; 116 (19.5%) of these were identified with adenomyosis [13]. Uteri were examined through routine histopathology using one LPF as the cut-off point for adenomyosis. There was a 20.5% coassociation of adenomyosis with myomas, 25.6% with genital prolapse, 17.8% with benign ovarian tumors, 13.6% with endometrial hyperplasia, 18.2% with cervical cancer, 16.1% with endometrial cancer, and 21.3% with ovarian cancer. The differences were not statistically significant. Almost half of the cases of adenomyosis had AUB. However, 54% of those with adenomyosis had fibroids as well, and the rest had other indications for surgery including cases of malignancy which may be linked to abnormal bleeding [13]. The presence of confounders raises doubts as to the relation between adenomyosis and AUB. This indicates similar incidence of adenomyosis irrespective of the indication of surgery.

In another large series by Parazzini et al. [14] of hysterectomies (n = 707), including 21.2% cases with adenomyosis, there was no relationship between adenomyosis and several menstrual characteristics, including polymenorrhea. The link to heavy cycles disappeared after adjustment for potential covariates. The frequency of adenomyosis was similar in women who had surgery for ovarian cysts (30.0%) and prolapse (31.4%) but was lower in women with fibroids and menorrhagia (14.8%) or genital cancer (24.7%); 48% of those with adenomyosis were postmenopausal. The study compiled participant’s subjective assessment of “lifetime menstrual pattern” and, using this parameter, found that there was no significant difference between those with and without adenomyosis. This indicates similar incidence of adenomyosis irrespective of the indication of surgery. Abnormal bleeding was the indication in 50.6% of 549 consecutive hysterectomies in the study by Bergholt et al. [15]. The incidence of adenomyosis varied from 10 to 18.2% depending on the cut-off criteria. There was no association between AUB and the presence of adenomyosis, but AUB was not defined, and a large proportion of cases were postmenopausal.

In a second study by Parazzini et al. [16], adenomyosis was present in 150 out of 707 consecutive hysterectomies. Menstrual histories, which were obtained prior to surgery, were categorized based on the following: (a) the “lifelong menstrual pattern” (three groups with cycle length ≤ 25, 26–30, and ≥31 days); (b) the number of days of bleeding per cycle (≤5 or >5); (c) the intensity of flow (“regular” and “heavy” bleeders). There was no difference in the incidence of heavy flow between the group with (58/146, 39.7%) or without (194/547, 35.4%) adenomyosis. In another study by Weiss et al. [17] of women who had a hysterectomy, AUB was present in 58% of those with, and 61% of those without, adenomyosis; adenomyosis was present in 44.8% of cases with, and 48% without, abnormal bleeding. This raised doubts as to the clinical significance of adenomyosis and whether it is a normal variant rather than a disease [17].

Özkan et al. [18] reviewed the records of 1680 patients who underwent a hysterectomy for benign indication and compared women with fibroids (n = 98, 5.8%) to those with adenomyosis (n = 106, 6.3%); overall, 40% were postmenopausal. Most (61%) of the group with adenomyosis and 48% of the group with fibroids were >50 years old. No indication is provided about associated pathology or the number of patients with concomitant fibroids and adenomyosis. Based on logistic regression, age, menometrorrhagia, and endometrial sampling were associated with adenomyosis. However, the incidence of menometrorrhagia in the adenomyosis group (35%) was lower than the incidence in the group with fibroids (43%), and assessment of menstrual bleeding was limited to classification into four groups (regular, oligomenorrhea, menometrorrhagia, and menopause). The inclusion of a significant proportion of postmenopausal women limits the interpretation of the results.

Pervez and Javed [19] reported adenomyosis in 296/861 (34.4%) uteri removed for AUB, excluding cases operated upon for cancer. In about half of these cases, adenomyosis was the sole pathology. The most common concomitant pathology was fibroids (50.6%). No detail was provided about the characteristics of AUB. This investigation also included postmenopausal women. The lack of a control group limits interpretation of the study. Naftalin et al. [20] analyzed a cohort of premenopausal women (n = 892) attending a gynecologic clinic for a range of indications. The majority had a pelvic ultrasound. There was no significant association between adenomyosis and menorrhagia diagnosed subjectively (binary response: yes or no) in multivariable analysis. When the severity of adenomyosis was assessed by counting the number of morphological features of adenomyosis on ultrasound, there was a significant (22%) increase in menstrual loss as assessed by PBAC for each additional feature of adenomyosis detected on ultrasound (OR 1.21 (95% CI: 1.04–1.40)). 

The study by Krentel and De Wilde [21] included two cohorts of routinely examined hysterectomies performed for benign indications. In the first cohort of hysterectomies for bleeding disorders and/or dysmenorrhea (n = 111) or prolapse (n = 42), adenomyosis was identified in 53 cases (34.6%). The diagnostic cut-off is not provided, but adenomyosis was found in 40.4% of cases with abnormal bleeding, with or without dysmenorrhea, compared to 23.8% with prolapse. The cohort had a high incidence of fibroids (75.5%). The indications for surgery in the second cohort of cases (n = 154) of laparoscopic subtotal hysterectomy were not provided. Adenomyosis was identified in 49.4%, fibroids in 68.2%, and both coexisted in 47.4% of cases. The report confirms the previously reported high incidence of adenomyosis in hysterectomies but does not provide any indication of the pattern or severity of bleeding. There was a higher incidence of adenomyosis in women with bleeding disorders in the first cohort but there was also a high incidence of concomitant fibroids.

The above studies provide a wide estimate of the incidence of adenomyosis. This may be attributable to the population studied or to the diagnostic criteria used to identify adenomyosis. The majority of the studies did not find a clear link between adenomyosis and bleeding. On the other hand, the studies did not involve comprehensive assessment of bleeding patterns. Indeed, many included diverse groups such as women with postmenopausal bleeding. A further challenge stems from the difficulty in accounting for concomitant pathology.

Bird et al. [22] identified adenomyosis in 123/200 (61.5%) of hysterectomy specimens: of these, adenomyosis was sub-basal within one low-power field (LPF) below the endometrium in 47 (38.5%) cases. Adenomyosis was considered the primary pathology in 92 cases and the sole pathology in 33 cases. No breakdown was provided of the symptoms or indications for surgery for the latter subgroup. Of the 92 women who had adenomyosis with no other significant pathology, 51.2% had menorrhagia, 10.9% had metrorrhagia, 2.2% had postmenopausal bleeding and 23.9% were asymptomatic. Of the 47 patients who had adenomyosis sub-basalis, 60% had significant menorrhagia compared to 19 of the 45 (42%) women who had deeper adenomyosis. No detail is provided about how symptoms or symptom severity were assessed. The indications for hysterectomy were not provided, but some patients in this group were receiving hormone treatments. The high incidence of reported cases is probably related to the classification of endometriosis sub-basalis as well as to increased number of histological sections examined per case.

In a small study by Goswami et al. [23], adenomyosis was found in 18/30 cases with menorrhagia who underwent preliminary endomyometrial biopsy using loop diathermy and in 11/30 controls with no menorrhagia. Menorrhagia was graded as 1+, 2+, and 3+. The depth of adenomyosis correlated with the severity of menorrhagia, but the indications for hysterectomy are not provided. 

Levgur et al. [24] included women (n = 111) who underwent hysterectomy for “benign uterine enlargement” with specimens weighing < 280 g. There were 17 cases with adenomyosis alone, 19 with adenomyosis and fibroids, 39 with fibroids alone, and 36 cases with neither fibroids nor adenomyosis. There was no significant difference between median numbers of adenomyotic foci in women with HMB and women without HMB. The depth of myometrial invasion was associated with HMB; this occurred in 36.8% of patients with deep foci and 13.3% with intermediate foci (*p* < 0.001). HMB was not associated with superficial foci. However, symptom distribution and indications for the hysterectomy were not provided. Sammour et al. [25] reported on 94 hysterectomy specimens containing adenomyosis. These were stratified based on the depth of adenomyosis within the myometrium into: Group A, up to 25% (n = 26); group B, between 26–50% (n = 36); group C, between 51–75% (n = 16), and group D, >75% (n = 16). The percentage of cases with menorrhagia in group A, B, C, and D was 61.5%, 61.1%, 68.7%, and 62.5%, respectively. Some women reported more than one primary symptom. There was no correlation between the spread of adenomyosis foci and HMB; 25 women with adenomyosis also had fibroids and 56% of this subgroup had menorrhagia.

Pinzauti et al. [26] compared nulliparous women aged 18–30 years who had regular cycles and no associated pathology based on the presence (n = 53) or absence (n = 103) of ultrasound features of diffuse adenomyosis. Most women with (79.2%) or without (69.9%) adenomyosis had regular cycles, comparable duration of bleeding (5.31 ± 1.41 and 5.58 ± 1.11 days, respectively), and amount of blood loss as assessed by the Pictorial Blood Loss Assessment Chart (PBAC) score (62 ± 30.7 and 57.54 ± 20.9). However, more women in the group with (n = 10/53) than in the group without (n = 3/103) adenomyosis had PBAC score ≥ 100. The proportion of women with features of adenomyosis on 2D-TVS was significantly higher amongst patients with HMB (i.e., PBAC ≥ 100 (10/13 (76.9%)) compared with those with PBAC < 100 (43/143 (30.1%)). Thus, whilst women with HMB were more likely to have diffuse adenomyosis, only 18.9% of women with adenomyosis had HMB. Bourdon et al. [27] recruited women who had MRI prior to surgery for benign gynecological conditions. Adenomyosis affecting the inner myometrium (n = 78) was more often associated with HMB, and adenomyosis of the outer myometrium (n = 109) was more frequently associated with endometriosis. The findings are necessarily influenced by the indication for surgery, which included pelvic pain, infertility, pelvic mass, and abnormal uterine bleeding. It is likely that the differences between the two groups were related to the indication for surgery. The study points to different symptoms based on the distribution of adenomyosis, with HMB linked to affection of the inner myometrium. Kilkku et al. [28] obtained a history from 212 women prior to hysterectomy for benign indications. Following surgery, 28 (13.2%) were diagnosed with adenomyosis. There was no symptom profile specific to adenomyosis, and no detail is provided about the reasons for the hysterectomy, how patients were selected, or the cut-off for diagnosis. Significant limitations hinder interpretations of the results.

Fraser et al. [29] evaluated menstrual blood loss in 55 women with a clinical history of menorrhagia and pelvic disease (40 women), or a coagulation disorder (15 women)c x. Three out of the five women with subjective menorrhagia and adenomyosis had objective menorrhagia, as measured by alkaline hematin. The measured blood loss in the adenomyosis group was 84.7 mL (SEM = 22.6) and was comparable to the group with associated endometriosis (83.8 ± 21.5 mL), but lower compared to the group with fibroids (171.7 ± 31.2 mL). Lack of a control group hinders understanding of the significance of adenomyosis in relation to AUB. Mobarakeh et al. [30] reported adenomyosis in 49/100 hysterectomies for AUB. In 28/49 cases, adenomyosis was associated with other pathology, mostly fibroids. Similar to the case in other studies, definition of AUB is not provided and the study included postmenopausal women. The study limitations do not enable drawing reliable conclusions of the link between adenomyosis and AUB.

Adenomyosis alone was found in 31/100 cases and concomitantly with fibroids in 6/100 cases and with hyperplasia in 11/100 cases in another series of hysterectomies for menorrhagia by Sawke et al. [31]. The cut-off point for adenomyosis and the menstrual history were not provided and the inclusion criteria were not adhered to. No reliable conclusions can be derived from the study because of its quality. Ates et al. [32] found no significant difference in terms of menorrhagia and metrorrhagia between women with fibroids who also had adenomyosis (n = 75) and women with fibroids only (n = 218) amongst women who underwent hysterectomy for benign disease. The study does not include details of how menstrual patterns were recorded, or the cut-off point for the diagnosis of adenomyosis. Thus, adenomyosis did not impact AUB symptoms.

Li et al. [33] compared the prevalence of lower urinary tract symptoms in 298 untreated symptomatic patients with adenomyosis diagnosed using ultrasound and 280 age-matched controls. It is not clear how patients with adenomyosis presented to health care leading to diagnosis. There was a higher incidence of menorrhagia in women with adenomyosis who had International Prostate Symptom Score (IPSS) ≥ 8 compared to women with IPSS score < 8 (51.1% vs. 30.8%, respectively), but the study was not set to examine the relation between adenomyosis and AUB.

Studies that attempted to assess the link between AUB and the extent or location of adenomyosis point to the possibility that HMB may be linked to adenomyosis subtypes. However, the findings have not been consistent and bleeding patterns have not been studied in detail.

Sabre et al. [34] investigated the distribution of causes of AUB according to PALM-COEIN classification in 390 women in a deprived community in New York City (68.2% Hispanic, 25.9% Black ethnicity). The most common type was the group with fibroids (AUB-L, 47.4%) whilst adenomyosis (AUB-A) was present in 14.1%. Ni et al. [35] classified 1065 women with AUB based on the FIGO PALM-COEIN classification using ultrasound and histopathology criteria. In this group 102 (9.6%) women were identified with adenomyosis, mostly (81.4%) diffuse disease.

In conclusion, the finding or adenomyosis in hysterectomy specimens of women with AUB could indicate a link, but hysterectomy is not inevitable in women with adenomyosis and/or AUB. A range of factors, including the presence of associated pathology and patient preference, are relevant to decision making, and it is important to consider the possible impact of the diagnosis of adenomyosis on the decision to proceed to investigations or surgery. Furthermore, AUB is not necessarily the sole, or the most significant, determinant of access to ultrasound or MRI, or to hysterectomy, in symptomatic women. Thus, the relation between adenomyosis and AUB cannot be ascertained with reference to diagnosis in selected subgroups, or in the absence of a structured approach to the assessment of menstrual bleeding.

**Table 1 biomolecules-14-00616-t001:** Studies that include case series.

Study	Intervention	Description	Findings
Studies with a focus on incidence			
Benson and Sneeden 1958 [10]	3276 premenopausal hysterectomies.	701 cases of adenomyosis, 245 asymptomatic, 112 as sole pathology. 56.6% of cases with adenomyosis had fibroids.	Menorrhagia was the sole symptom in 38.4% of cases with only adenomyosis and in 28.5% of cases in which there were also associated pathologies.
Molitor 1971 [11]	3207 hysterectomy specimens.	281 cases of adenomyosis, 81 asymptomatic, 38.5% of cases with adenomyosis had fibroids.	In 58% of symptomatic cases, adenomyosis was the sole pathology. In cases of adenomyosis of the inner third of myometrium, 15.8% were symptomatic. In cases affecting the inner 2/3 or the whole myometrial thickness, 65% and 82.7% respectively were symptomatic.
Vercellini et al. (1995) [12]	1334 consecutive hysterectomies.	332 (24.9%) had adenomyosis. The incidence was not statistically different in case of fibroids and menorrhagia, prolapse, ovarian cysts, cervical cancer, endometrial cancer, ovarian cancer, or other indications.	Adenomyosis in 23.3% of cases with fibroids and menorrhagia, 25.7% of cases with prolapse, 21.4% of cases with ovarian cysts, 19% of cases with cervical cancer.
Vavilis et al. (1997) [13]	Retrospective review of 594 hysterectomies.	Adenomyosis in 116 (19.5%). Almost half cases of adenomyosis had AUB, 54% of adenomyosis had fibroids.	AUB in 54/120 cases with and 66/354 cases without adenomyosis.
Parazzini et al. (1997) [14]	707 hysterectomies.	372 had hysterectomy for fibroids and/or menorrhagia, 140 for prolapse, 100 for ovarian cysts. 150 cases of adenomyosis.	Adenomyosis was present in 30% of cases of ovarian cysts, 31.4% in cases with prolapse, and 14.8% in women with fibroids and menorrhagia.
Bergholt et al. (2001) [15]	549 consecutive hysterectomies.	Incidence of adenomyosis 10–18.2% depending on cut-off point for diagnosis.	278 women had bleeding disorders and 271 did not have bleeding disorders.
Parazzini et al. (2009) [16]	820 consecutive hysterectomies.	231 had adenomyosis. 143 (28.2%) had adenomyosis in cases with menorrhagia compared to 69 (28.5%) in cases with prolapse.	13% of those with adenomyosis and 11.8% without adenomyosis (n = 558) had irregular cycles.
Weiss et al. (2009) [17]	137 women who had hysterectomy.	48% had adenomyosis. Frequency of presenting symptoms was similar in those with and without adenomyosis.	Abnormal bleeding was present in 27% of those with and 33% of those without adenomyosis.
Özkan et al. (2011) [18]	Record review of 1680 cases of benign hysterectomies.	106 cases had adenomyosis.	35% of those with adenomyosis had menometrorrhagia, 49% were menopausal, 13% had regular cycles and 3% had oligomenorrhea.
Pervez and Javed (2013) [19]	Retrospective study of hysterectomies (n = 861).	All had abnormal bleeding.	296 had adenomyosis, including 150 who also had fibroids and 16 who had polyps.
Naftalin et al. (2014) [20]	Prospective study of 714 consecutive women ascending clinic.	157 had adenomyosis on ultrasound. No significant association between adenomyosis and menorrhagia as a binary outcome.	93/157 (46%) women with adenomyosis and 256/558 (46%) women without adenomyosis had menorrhagia.
Krentel and De Wilde (2022) [21]	Retrospective series of hysterectomies (n = 307) examined in two cohorts.	42% incidence of adenomyosis. Highest incidence of adenomyosis (59.3%) in women with bleeding and dysmenorrhea.	74/153 (48.4%) underwent hysterectomy for bleeding disorders, 25/74 (33.8%) of these had adenomyosis.
Studies with a focus on disease distribution			
Bird et al. (1972) [22]	200 hysterectomy specimens.	123 cases of adenomyosis. Adenomyosis was the primary pathology in 92 cases and the sole pathology in 33 cases.	Menorrhagia was present in 28 of the 47 cases with adenomyosis sub-basalis, and in 19 of the 45 cases with grade II and III adenomyosis.
Goswami et al. (1998) [23]	83 participants.	53 patients with menorrhagia and adenomyosis diagnosed by endomyometrial biopsy vs. 30 hysterectomies for other indications.	Adenomyosis in 60% (n = 18) in the endomyometrial biopsy group and 33.3% in controls.
Levgur et al. 2000 [24]	111 hysterectomies for benign uterine enlargement.	17 cases of adenomyosis alone and 19 cases of adenomyosis and fibroids. Deeps adenomyosis foci linked to heavy menstrual bleeding.	The median number of adenomyosis foci in women with or without menorrhagia was 7. Menorrhagia was present in 36.8% in deep, and 13.3% of intermediate depths adenomyosis.
Sammour et al. 2005 [25]	94 hysterectomy specimens containing adenomyosis.	Stra9fied based on the depth of endometrium within the myometrium.	Group A, B, C, D glands within 25%; 26–50%; 51–75%, and >75% of myometrial thickness. Menorrhagia in 61.5%, 61.1%, 68.7% and 62.5% of the 4 groups, respectively.
Pinzauti et al. (2014) [26]	53 women with ultrasound diagnosed adenomyosis and 103 unaffected.	A significant relation between the number of ultrasound features of adenomyosis and PBAC scores.	PBAC score 62 ± 30.36 and 57.54 ± 20.85 in women with (n = 53) and without (n = 103) diffuse adenomyosis.
Bourdon et al. (2021) [27]	Adenomyosis of the inner (n = 78) and outer (n = 109) myometrium.	Symptom comparison. Internal adenomyosis linked to HMB.	Heavy menstrual bleeding was more common (n = 62, 80%) in women with internal compared to women with external (n = 58, 53.2%) adenomyosis.
Other studies			
Kilkku et al. (1984) [28]	212 women who had hysterectomy.	28 had adenomyosis.	The average duration of bleeding was 6 days (range 3–11) in the adenomyosis group and 6 days (range 4–10) in those without adenomyosis. In the control group, it was 6 days.
Fraser et al. (1986) [29]	55 women with menorrhagia, pelvic disease, or coagulation disorder.	MBL in 5 cases with adenomyosis was 84.7 ± 22.6 mL.	Compared to women with fibroids (171.7 ± 31.2 mL), myometrial hyperplasia (162.6 mL), endometriosis (83.8 ± 21.5 mL).
Mobarakeh et al. (2012) [30]	100 cases of AUB who underwent hysterectomy.	49% had adenomyosis, either alone (21%), with fibroids (21%) or with other pathology (7%).	Age range 21–75 years, 7% were >60 years old.
Sawke et al. (2015) [31]	100 uteri removed for heavy periods.	Inclusion criteria was women ages 30–50 year, but 14 cases were 21–30 years old and 15 were 51–60 years old.	48 had adenomyosis. Of these, 11 also had endometrial hyperplasia and 6 had fibroids.
Ates et al. (2016) [32]	75 women with fibroids and adenomyosis vs. 218 with fibroids.	No difference in menorrhagia between the two groups.	Menorrhagia and metrorrhagia in 25 (33.3%) and 21 (28%), respectively, in women with adenomyosis and fibroids cf. 88 (40.4%) and 70 (32.1%) of women with fibroids only.
Li et al. (2017) [33]	Women with adenomyosis (n = 298) and age matched controls (n = 280).	In the adenomyosis group, women with a high international prostate symptom (IPSS) score had a higher incidence (51.1%) of menorrhagia compared to those with a low IPSS score (30.8%).	Patients with heavy menstrual bleeding experienced more than a twofold increase in the odds of IPSS total ≥ 8. The study does not provide the incidence of menorrhagia in the control group.
Studies using FIGO classification			
Sabre et al. (2021) [34]	390 women with AUB.	55 had adenomyosis.	The most common AUB type was AUB-L (n = 185, 47.4%), followed AUB-P (n = 100, 25.6%), AUB-A (n = 55, 14.1%).
Ni et al. (2022) [35]	Retrospective classification of women (n = 1065).	102 (9.58%) had AUB-A based on FIGO classification.	The most common classification was AUB-L (45.73%), AUB-P (16.53%), AUB-A (9.58%), or AUB-M (2.91%).

#### Studies That Addressed FIGO Classification

As mentioned above, only two studies in this group included a reference to the FIGO PALM-COEIN classification. Sabre et al. [34] was a descriptive cross-sectional study of women attending an outpatient clinic in an area of deprivation. Women with AUB were classified according to PALM-COEIN system based on one or more of history, clinical examination, ultrasound, hysteroscopy, biopsy, and blood tests. The diagnosis of AUB-A was made in 14.1% of cases. There is no indication that any participant underwent MRI, and it is unclear how many women in the cohort underwent ultrasound. There is no indication of concomitant pathology. Furthermore, the pattern of bleeding is not provided. Ni et al. [35] included 1065 women presenting with AUB. The study included women with a range of AUB, including frequent menstruation, menorrhagia, prolonged menstrual period, sparse menstrual period, and intermenstrual bleeding, amongst others. Histopathology (including curettage, hysteroscopic resection, or hysterectomy) and ultrasound results were reviewed retrospectively and used for the purpose of PALM-COEIN classification, and 9.6% (n = 102) were assigned as AUB-A. Adenomyosis was the sole pathology in 5.54% (n = 59) of cases, but was present together with fibroids in 2.4% (n = 25) and less frequently in cases with malignancy, polyps, or multiple pathologies. It is not possible from this study to assess if adenomyosis is relevant to any particular bleeding pattern.

### 3.2. Studies That Addressed the Response to Hormonal Preparations

These investigations enrolled patients who presented with menstrual abnormalities amongst other symptoms, mostly dysmenorrhea. Adenomyosis was detected by ultrasound or MRI. In most cases, detailed menstrual history or how that was affected following treatment was not provided beyond broad reference to excessive loss or drop in hemoglobin.

#### 3.2.1. Dienogest (Table 2)

In a study by Nagata et al. [36], 51 patients with adenomyosis who received dienogest identified uterine bleeding as the most common reason for treatment discontinuation. However, the study did not provide the bleeding pattern before or after dienogest. Similarly, Hirata et al. [37] reported worsening anemia because of metrorrhagia following dienogest administration in a small study of 17 women with symptomatic adenomyosis. The study by Fawzy and Mesbah [38] included 41 women with adenomyosis who received dienogest (n = 22) or triptorelin acetate (n = 19). Participants were symptomatic with pain and menorrhagia. Dienogest reduced blood loss in the majority of participants, but persistent, heavy, prolonged, or irregular bleeding were the main reason for discontinuation.

Hassanin et al. [39] included 110 women with dysmenorrhea and ultrasound diagnosed adenomyosis who may or may not have had menorrhagia and were randomly treated with dienogest or combined oral contraceptives (COCs). An unspecified number of participants were IUD users. The study reported that treatment reduced average number of days of bleeding and number of sanitary pads used and increased bleed-free days and days of spotting. Because of the design of the studies, it is not possible to draw any conclusion about the relation between bleeding and adenomyosis. Kobayashi [40] identified the presence of internal adenomyosis as a risk factor for abnormal bleeding in women receiving dienogest.

**Table 2 biomolecules-14-00616-t002:** Studies that addressed response to dienogest and other hormone treatments.

Study	Intervention	Description	Findings
Nagata et al. (2012) [36]	Retrospective study, 51 women with adenomyosis treated with dienogest.	Higher risk of discontinuation of dienogest due to bleeding if younger, have anemia before treatment, or have mildly suppressed or unsuppressed estradiol.	Discontinuation rate 11 (21.6%): 8 (15.7%) due to bleeding.
Hirata et al. (2014) [37]	17 women with symptomatic adenomyosis treated with dienogest.	5 had concomitant fibroids.	5 women experienced worsening of anemia due to heavy bleeding on dienogest. One discontinued because of severe bleeding.
Fawzy and Mesbah (2015) [38]	22 women treated with dienogest compared to 19 treated with triptorelin.	All participants described as having menorrhagia (100%). Triptorelin was more effective in controlling menorrhagia.	At 6 months, bleeding was heavy in 5/19 (26.3%) of the dienogest group but in none of the triptorelin group. 21.1% of the dienogest and 94.4% of the triptorelin groups were amenorrhoeic.
Hassanin et al. (2021) [39]	RCT comparing dienogest vs. combined pill in adenomyosis.	Bleeding pattern improved greatly in both groups.	Number of days of bleeding reduced in the dienogest group from 9.55 ± 2.17 at baseline to 5.54 ± 2.47 at 6 months in the dienogest group (n = 48) and from 10.05 ± 1.94 to 7.80 ± 1.38 in the combined oral contraception group (n = 49).
Kobayashi (2023) [40]	Review article of dienogest for adenomyosis.	Bleeding as a complication of dienogest use may be related to age, diffuse disease, severe dysmenorrhea, elevated CA125, low Hb.	
Niu et al. (2021) [41]	Prospective study of etonogestrel in adenomyosis.	100 women with adenomyosis or endometriosis: menstrual loss reduced significantly compared to baseline.	74 completed the study. Menstrual volume was reduced from baseline by 40.69% ± 30.92.
Garcia et al. (2018) [42]	41 patients with adenomyosis and fibroids vs. 122 patients with fibroids only treated with ulipristal acetate.	Significant improvement in symptoms.	More women in the adenomyosis and fibroid group (90.24%) had amenorrhea compared to the fibroid only group (73.68%).
Capmas et al. (2021) [43]	RCT ulipristal acetate (n = 30) vs. placebo (n = 10).	AUB and adenomyosis (diagnosed by imaging). Reduction in bleeding score during treatment.	No woman in the placebo group versus 95.2 % in the ulipristal acetate group had a PBAC score < 75 during the 28 days following the 12-week treatment.
Goncalves-Henriques et al. (2022) [44]	Systematic review of ulipristal acetate use.	Ulipristal acetate results in partial or complete remission of AUB in women with adenomyosis.	
Sun et al. (2023) [45]	Prospective study (n = 139) comparing monthly (n = 69) to trimonthly (n = 70) goserelin.	significant decrease in the dysmenorrhea (NRS) score, uterine volume, and induced amenorrhea.	68 and 62 patients in the goserelin 10.8 mg and 3.6 mg groups, respectively, completed the study. All had amenorrhea at week 12 except for one participant receiving monthly injections.

#### 3.2.2. Etonogestrel Implants

Niu et al. [41] evaluated the effect of a subcutaneous etonogestrel implant on menstrual flow in 100 women with dysmenorrhea who were diagnosed with adenomyosis (n = 56), or endometriosis (n = 44). Adenomyosis was diagnosed by ultrasound. There was a reduction in menstrual flow following etonogestrel, but whilst the relation between adenomyosis and menstrual patterns cannot be ascertained from this study, it is notable that only 26 of all participants had menorrhagia.

#### 3.2.3. Ulipristal Acetate

Gracia et al. [42] evaluated the response to ulipristal acetate (UPA) of a group of women with symptomatic fibroids. All women were referred because of AUB and pelvic pain. These were divided into a group with fibroids only (n = 122) and a group with fibroids and adenomyosis (n = 42). The incidence of AUB was comparable in both groups. UPA treatment was associated with a higher rate of amenorrhea in women who had concomitant adenomyosis. Capmas et al. [43] recruited 40 women with AUB, with or without pelvic pain, who had adenomyosis into a placebo-controlled trial of UPA. All women had a PBAC score of >100 at baseline and adenomyosis was diagnosed by ultrasound or MRI, but no detail is provided of the bleeding pattern, or of those patients screened for eligibility.

The review by Goncalves-Henriques et al. [44] concluded that UPA induces partial or complete remission of AUB in adenomyosis.

#### 3.2.4. Goserelin Acetate

Sun et al. [45] compared the efficacy and safety of goserelin, either by monthly or trimonthly administration, in 139 women with symptomatic adenomyosis. Women were included if they had dysmenorrhea and/or menorrhagia and had been diagnosed with adenomyosis by ultrasound. Women with concomitant fibroids were excluded. Given its mode of action, it is not surprising that goserelin was effective in inducing amenorrhea in the vast majority of cases, but the study design does not enable examination of the link between adenomyosis and menstrual abnormalities.

#### 3.2.5. The Levonorgestrel-Releasing Intrauterine System (Table 3)

A number of investigations evaluated the effect of the levonorgestrel-releasing intrauterine system (LNG-IUS) in women with menorrhagia and adenomyosis diagnosed with ultrasound or MRI. Some of these studies used PBAC charts.

The study by Fedele et al. [46] recruited women with recurrent menorrhagia who were diagnosed with adenomyosis on ultrasound, but no detail is provided of the population screened for eligibility. Similarly, Cho et al. [47] did not provide information of the population from which the 47 participants were recruited. Ozdegirmenci et al. [48] recruited patients with menorrhagia, with or without dysmenorrhea or dyspareunia. These underwent screening to identify those with adenomyosis, first by ultrasound and then by MRI. Those with submucous fibroids, or with intramural or subserous fibroids >2 cm, were not eligible for enrolment. In the subgroup in the hysterectomy arm of the RCT, adenomyosis alone was confirmed in 65.5% of the cases; adenomyosis and fibroid in 9.4%; 25% did not have histological adenomyosis. Desai et al. [49] recruited 40 perimenopausal women with a range of benign lesions of the uterus associated to menorrhagia. Blood loss was assessed by PBAC, and participants were followed-up for 1 year. Following insertion of the device, a majority initially had spotting, followed by infrequent or scanty menstruation, or amenorrhea.

In the study by Kelekci et al. [50], the response to LNG-IUS was compared in women who did have adenomyosis on ultrasound and in those who were seeking contraception and were asymptomatic. Those with adenomyosis had longer duration of bleeding (6.67 ± 1.66 days) compared to those who had LNG-IUS (5.21 ± 1.13 days) or had copper devices (4.68 ± 1.02 days) amongst those seeking contraception. However, the groups were not matched and there is no indication as to the clinical presentation of the group with adenomyosis.

**Table 3 biomolecules-14-00616-t003:** Studies that addressed response to LNG-IUS.

Study	Intervention	Description	Finding
Fedele et al. (1997) [46]	25 women with menorrhagia and adenomyosis had LNG-IUS.	One year of follow-up completed by 23 women.	2 had amenorrhea, 3 oligomenorrhea, 2 spotting, and 16 regular flow; none had menorrhagia.
Cho et al. (2008) [47]	47 women with adenomyosis.	Menstrual loss as measured by PBAC score, reduced markedly at 6 and 24 months after LNG-IUS insertion.	PBAC score at baseline = 201.43 ± 100.17, at 6 months = 10.95 ± 7.34, at 24 months = 9.02 ± 5.43.
Ozdegirmenci et al. (2011) [48]	86 women with adenomyosis randomized to LNG-IUS or hysterectomy.	43 per group.	Ten (23.8%) and 22 (51.4%) patients in the LNG- IUS group were amenorrhoeic at 6 month and 12 months, respectively, whilst 18 (42.8%) and 15 (35.7%) patients had oligomenorrhea.
Desai (2012) [49]	40 women with menorrhagia had LNG-IUS.	4 women had adenomyosis. LNG-IUS proved effective.	LNG-IUS proved effective. No specific information about the adenomyosis cohort.
Kelekci et al. (2012) [50]	LNG-IUS inserted in 23 women with ultrasound diagnosed adenomyosis.	Improved duration of bleeding and Hb levels.	Days of bleeding reduced from 6.64 ± 1.63 days at baseline to 3.68 ± 1.94 at 12 months.
Uysal et al. (2013) [51]	42 women with adenomyosis faced with LNG-IUS.	Retrospective study.	Amenorrhea, oligomenorrhoea, spotting, and regular menstrual flow occurred in 4, 3, 8, and 27 women, respectively, at month 6, and in 4, 3, 5 and 30 women, respectively, at month 12.
Yang et al. (2022) [52]	Comparative study of a modified (frameless) LNG-IUS and LNH-IUS (n = 47 each arm).	Both were effective in reducing menstrual blood loss and improving hemoglobin.	PBAC for the Modified LNG-IUS group was 501.1 ± 202.8 at baseline and 23.0 ± 15 at 12 months. The corresponding figures for the GnRH + LNG-IUS were 450.2 ± 210.7 and 23.7 ± 15.6, respectively.
Ishizawa et al. (2023) [53]	Comparison of LNG-IUS in early vs. advanced adenomyosis.	The menorrhagia mul9-acribute scale was used. Adenomyosis diagnosed by MRI.	LNG-IUS was more effective in menorrhagia specific HR-QOL and in improving Hb in incipient * adenomyosis compared to advanced disease.

* Incipient adenomyosis: lesion localized in the inner or outer side of the myometrium, and the healthy myometrium is retained in the opposite side. In advanced adenomyosis, lesion involves the whole myometrial thickness.

Our search identified other studies that did not include information about pretreatment menstrual patterns, or the presenting symptoms of women with adenomyosis [51], did not detail how adenomyosis was diagnosed, or include detail for menstrual history [52], or had a high percentage of patients with concomitant fibroids [53].

### 3.3. Studies That Addressed Response to Surgery and/or Ablation (Table 4)

There are reports of successful outcome of ablation in the presence of adenomyosis and two reported cases of adenomyosis first identified following ablation, but it is not certain if these represent a newly developed or previously unidentified disease [54]. In an early study of endometrial ablation, McCausland and McCausland [55] reported that successful outcome of ablation is inversely related to the depth of endometrial ablation. The study included 18 women with endometrial depth of ≥2 mm and 32 women with lower depth of myometrial involvement. The authors considered 2 mm as the cut-off point to diagnosis of adenomyosis. In this study, the depth of invasion was addressed through an endomyometrial biopsy taken from the posterior wall of the uterus using a loop electrode. The figures provided indicate heavier blood loss in women with deeper myometrial affection, but there is no indication as to how representative the biopsy sample was for the rest of the uterus. Chan et al. [56] reported a case of successful reduction in menstrual flow following balloon endometrial thermoablation.

Similar challenges are encountered in studies that addressed the response to uterine artery embolization (UAE) [57,58,59,60,61,62,63] as they cannot address the question of whether adenomyosis is the cause of abnormal bleeding. These studies included cases of focal or diffuse adenomyosis, as well as cases with concomitant fibroids. In the study by Kang et al. [61], women (n = 37) with menorrhagia and adenomyosis underwent uterine artery occlusion and partial resection of the adenomyosis-containing uteri. Most (n = 29) women had focal adenomyosis, 8 had diffuse disease, 9 had concomitant fibroids. Uterine artery occlusion, however, is not specifically targeted to adenomyosis.

Osada et al. [64] reported on radical resection of adenomyosis in 104 cases who had adenomyosis affecting >80% of the anterior and posterior uterine walls. Women had significant uterine enlargement, and the weight of excised tissue was 292.6 ± 254.1 g. Menstrual blood loss, although assessed using a nonvalidated visual analogue score, showed postoperative improvement. Another surgical approach to women with symptoms linked to adenomyosis is laparoscopic uterine artery occlusion [65], which has been undertaken in conjunction with resection of adenomyosis.

There is information on the effect of high-intensity focused ultrasound (HIFU). Zhang et al. [66] treated 120 women with focal, and 82 with diffuse, adenomyosis using HIFU; 139 women had menorrhagia. The average improvement in the menorrhagia score was similar in the groups with focal or diffuse adenomyosis. In the study by Shui et al. [67], women (n = 224) were followed-up after undergoing HIFU for adenomyosis. Menstrual symptoms were recorded on a scale of 1 to 5. The percentage of women who experienced normal cycles, moderately large, large, very large, and extremely large loss was 51.4%, 13.8%, 24.5%, 9.4%, and 0.9%, respectively. There is no information on whether this correlated with the extent of adenomyosis, although not all those affected with adenomyosis had troublesome periods and the percentage with extremely heavy loss was small. Zhou et al. [68] studied the outcome of UAE for adenomyosis in relation to lesion vascularity: whether the uterus was equally supplied by both uterine arteries and whether the lesion was hyper-, iso-, or hypo-vascular. The findings suggest a role for lesion vascularity as a predictor of outcome. Pyra et al. [69], in a small group (n = 12) of women with adenomyosis who had dysmenorrhea, menorrhagia, and problems with urination, found that menstrual bleeding was reduced following endovascular embolization. No conclusion can be drawn regarding the link between adenomyosis and symptoms.

**Table 4 biomolecules-14-00616-t004:** Studies that addressed response to surgery and/or ablation.

Study	Intervention	Description	Findings
Yuen (1995) [54]	A report of two cases of rollerball ablation.	Failed treatment and required a hysterectomy.	Dysmenorrhea and uterine enlargement were diagnosed after ablation. Adenomyosis was diagnosed at subsequent hysterectomy.
McCausland and McCausland (1996) [55]	Review of the effectiveness of endometrial ablation in adenomyosis.	Literature review.	Superficial (<2.5 mm), but not deep adenomyosis yield good results from the ablation.
Chan et al. (2001) [56]	Case report: one woman with heavy periods underwent endometrial ablation.	Reduced blood loss.	Reduced blood loss from 96.9 mL to 37.57 mL at 6 months.
Siskin et al. (2001) [57]	15 patients with menorrhagia and adenomyosis underwent UAE.	5 had diffuse and 1 had focal adenomyosis with no fibroids. 9 had concomitant fibroids.	Significant improvement in symptoms in 12 cases; 1 had amenorrhea 3 months after surgery.
Kim et al. (2004) [58]	43 women with adenomyosis underwent UAE.	Significant improvement in menorrhagia (95%).	Improvement in 38 of the 40 patients who had complained of menorrhagia after 3.5 months (range 1–8 months).
Chen et al. (2006) [59]	168 women with adenomyosis who underwent UAE and completed follow-up.	74 women had normal bleeding before UAE. Bleeding did not change in 77% (57/74), became lighter in 20.3% (15/74) and 2 developed amenorrhea.	93 had menorrhagia before surgery: 78 (83.9%) returned to normal and 10 (10.8%) developed oligomenorrhea.
Kim et al. (2007) [60]	Retrospective study of 54 women with adenomyosis who had UAE with 3 year follow-up.	12 lost to follow-up. 4 immediate failures, 19 had relapse. Mean time of symptom recurrence 17.3 months.	Significant relief of symptoms in the short term. Aber a mean follow-up of 4.9 years, 19 showed improvement.
Kang et al. (2009) [61]	37 symptomatic patients had uterine artery occlusion combined with partial resection of adenomyosis.	Significant improvement in pictorial blood loss assessment at 1, 6, and 12 months.	Menorrhagia score at baseline, 1, 6, and 12 months was 158, 58, 56, and 59, respectively.
Bratby and Walker (2009) [62]	27 cases of adenomyosis who underwent uterine artery embolization.	14 also had fibroids.	Menorrhagia improved in 79% of cases at 12 months. Of those followed up for a year (n = 11), 5 (45.5%) reported deterioration.
Kim (2011) [63]	40 patients with adenomyosis underwent UAE.	No recurrence of menorrhagia in women with complete necrosis of adenomyosis.	33 had complete necrosis of adenomyosis.
Osada et al. (2011) [64]	104 women underwent adenomyomectomy.	Dramatic reduction in hypermenorrhea.	VAS for hypermenorrhea: 3.27 ± 2.17, 2.89 ± 1.77, 2.63 ± 1.3, and 2.87 ± 1.77 at 3, 6, 12, and 24 months after surgery.
Liu et al. (2014) [65]	182 with symptomatic adenomyosis underwent laparoscopic bilateral uterine artery occlusion and resection of adenomyosis.	Improved mean menorrhagia scores at 3, 6, 12, and 36 months.	Pictoral blood loss assessment scores reduced from 146 (95% CI 128–235) at baseline to 58 (95% CI 29–78) at 36 months.
Zhang et al. (2014) [66]	Retrospective review of 202 women who underwent ultrasound-guided HIFU (120 focal and 82 diffuse adenomyosis).	Significantly reduced menorrhagia score.	23 patients completed 23 months of follow-up. Their baseline score improved from 3.0 ± 0.5 to 1.2 ± 0.4.
Shui et al. (2015) [67]	Follow-up of 224 women who underwent ultrasound-guided high-intensity focused ultrasound.	Significant reduction in blood loss in (79.8%, 80.7%, and 78.9% at 3months, 1 and 2 years). Menstrual volume 5-point score (1–5 based on severity).	Menstrual volume score reduced from 2.9 ± 0.8 at baseline to 1.6 ± 1.0 at 24 months.
Zhou et al. (2016) [68]	Retrospective study of 264 patients who underwent UAE.	4 failed procedures, 1 died from embolism, 7 ovarian failure.	Menorrhagia improved in 70.9% at 12 months and 68.8% at 5 years. Improvement rate was higher in cases of hyper vascular adenomyosis.
Pyra et al. (2022) [69]	Preliminary study of endovascular embolization (n = 12).	5 menopausal within 12 months. Significant reduction in bleeding and pain (one case of pain recurrence).	All patients had adenomyosis alone or with fibroids. All had reduction in menstrual flow following embolization.
Liu et al. (2021) [70]	Review of effectiveness of image-guided thermal ablation of adenomyosis.	Meta-analysis (high-intensity focused ultrasound (HIFU), percutaneous microwave ablation (PMWA) and radiofrequency ablation (RFA). The Symptom Severity Scores (SSS) was used to assess changes in menstrual loss.	Standardized mean difference (SMD) from baseline for HIFA was 2.14 (95% CI, 1.56–2.72) and for PMWA was 1.45 (95% CI, 0.47–2.44). For RFA, the SMD was 2.15 (95% CI, 0.03–4.33).

A review of the outcomes of image-guided thermal ablation for symptomatic adenomyosis reported on average changes to the menorrhagia severity scores, but it is not possible from this report to ascertain a relation between menorrhagia severity and extent of adenomyosis [70]. 

## 4. Discussion

As histopathology is the gold standard for the diagnosis of adenomyosis, agreement on the cut-off point for diagnosis is relevant to standardization of imaging-based criteria. However, as alluded to above, there is no agreement on the cut-off point at which irregularity at the endometrial–myometrial interface can be considered pathological [71]. Indeed, studies that employed histological sectioning do vary in the adopted definition, which remains arbitrary. This has been shown to affect estimates of disease prevalence [7,72]. The dividing line between the normal and the pathological needs to take into account the clinical significance of the presence of endometrium within the myometrium and when this becomes symptomatic, i.e., it represents a disease. Many authors considered a high prevalence of adenomyosis in hysterectomy samples as sufficient evidence that the presence of ectopic endometrium is the underlying reason for the symptoms and, as such, constitutes a disease, but others have pointed to instances where ectopic endometrium has been identified in comparable proportions in cases that are either asymptomatic or where uteri were removed for indications other than the “classic” symptoms of AUB and dysmenorrhea [8]. In this review, we collated studies that have relevance to AUB as a symptom that has traditionally been linked to adenomyosis.

Determining the relation between adenomyosis and symptoms, including AUB, requires the identification of cases with adenomyosis either amongst those with the index symptom(s) or at a population level. In the past, this was largely hampered by the lack of a widely available, reliable method for noninvasive diagnosis. Over the last two decades, ultrasound and MRI have become more widely adopted, but these are not without their own difficulties; prominent among them is the question of reproducibility, and the correlation between imaging features and histological subclassifications of adenomyosis [7,22]. There are indications that these subclassifications themselves, including the notions of sub-basal adenomyosis and junctional zone hyperplasia, the depth of invasion, disease spread, and lesion distribution and density, are relevant to symptoms [22]. Additional levels of complexity arise because other conditions, such as fibroids, commonly coexist with adenomyosis and because symptoms, such as pain and abnormalities of menstruation, commonly occur in combination.

The vast majority of the investigations cited above used routine histological assessment and included retrospective reports which may or may not have sought to exclude the disease. Most reported retrospective case series did not specify the diagnostic criteria for adenomyosis or the indications for hysterectomy. Stating the indication for surgery is critical but is also complex because of the multiple factors that underpin clinical decisions. Some, but not all, of these factors are related to symptoms. However, the relative importance of individual symptoms is difficult to ascertain. Whilst none of the studies identified in this review contained a detailed analysis of menstrual patterns, some used nonvalidated tests, global generalizations, or oversimplifications of a very complex topic.

The lack of agreement on the diagnostic criteria for minimal or sub-basal adenomyosis is necessarily linked to the limited understanding of its clinical significance and associated symptoms. Nevertheless, even if symptoms, individually or in combination, may be poorly understood, it can probably be agreed that the presence of endometrial glands deep within the myometrium is a structural aberration. The question is less clear for sub-basal lesions, yet these have been liked to symptoms in one study [22]. There is an increasing role of modern imaging in enabling noninvasive diagnosis, but its role remains hampered by the lack of studies that map adenomyosis within the uterus, or correlate ultrasound features with disease distribution and severity. There also remain challenges in understanding the significance of the junctional zone seen using different imaging techniques [73]. Given this, it is hardly surprising that the incidence of adenomyosis in the general population is poorly understood. A small study reported adenomyosis in 54% of 54 autopsies [74]. In the investigation of Hauth et al. [74] using MRI, adenomyosis was present in 12% of healthy women. The wide difference in the incidence of adenomyosis in various studies is difficult to explain but may be a factor of the diagnostic criteria used and/or of the study population [72].

The FIGO expert committee opined that the relation between adenomyosis and the genesis of AUB is unclear, and it highlighted the need for extensive additional research [5,6]. The approach adopted by the FIGO committee was a modified RAND/UCLA Delphi process, and the committee referred to methods of diagnosis and to an article that argued that adenomyosis is a nonpathological variant [17]. They included a category for adenomyosis in their classification (AUB-A), with the aim of standardizing the classification of the causes of AUB. In support of including AUB-A is the existence of sonographic and MRI-based diagnostic criteria. The expert group argued that ultrasound diagnosis should include the agreed minimum sonographic criteria, and a distinction between diffuse and focal (or multifocal) disease. They also proposed the inclusion of a metric indicating the volume, or extent of the disease [5,6].

Investigating the etiology of AUB is hampered by confusing and inconsistent use of nomenclature and by the lack of standardization [5]. The FIGO classification can thus been viewed as an effort to support good quality research and comparisons between studies, but its utility more than a decade later remains questionable. Only few studies that addressed adenomyosis and bleeding contained reference to PALM-COEIN classification, and these used it only in the context of tabulating case series into the various categories. Arguably, rather than triggering curiosity into the significance of adenomyosis, the classification may have merely enforced the impression that the relation between adenomyosis and AUB is well established. On the other hand, the notion of AUB itself has been used to include conditions as diverse as HMB, erratic bleeding, intermenstrual bleeding, and bleeding after the menopause.

The FIGO PALM-COEIN classification includes adenomyosis as a structural cause for AUB, which includes HMB [5]. Traditionally, HMB was defined as menstrual blood loss (MBL) greater than 80 mL [75]. The mechanisms of control of menstrual blood loss involve endometrial COX-2, PGE_2_ and PGF_2α_, local hypoxia, and the activation of hypoxia inducible factor 1α (HIF-1α) which establishes a microenvironment conducive to tissue repair and angiogenesis [76]. Disruption of these signaling pathways is linked to HMB [76,77]. Adenomyosis has been linked to mechanisms that can lead to increase blood loss, resulting from increased lesional fibrosis [78,79,80,81] and tissue stiffness or rigidity [82]. Endometriotic stromal cells cultured in stiffer substrates demonstrate downregulation of COX-2 and E-series receptor type 2 (EP2) and EP4 [83].

Profibrotic molecules secreted in lesions may permeate into their neighboring junctional zone (JZ) and the eutopic endometrium, causing increased endometrial stiffness. This would lead to attenuation of local PGE_2_ signaling with reduced expression of COX-2, EP2, and EP4, and, thus, reduced PGE_2_ [3,83], which may result in suppression of HIF-1α and, thus, HMB [3,83]. Increased endometrial fibrosis leads to reduced expression of histone deacetylase 3 (HDAC3), leading to dampened inflammation and impaired endometria repair, and, thus, HMB [84]. Patients with adenomyosis who complained of excessive MBL have lower glycolysis than those with moderate to heavy MBL [85]. Boosting glycolysis by pharmacological means can alleviate the amount of simulated MBL in mice with induced adenomyosis without exacerbating adenomyosis [86].

These findings are consistent with reports that different subtypes of adenomyosis have different symptoms [22,27,87]. Considering this in the context of the findings in this review emphasizes the need for better characterization of the clinical profile of adenomyosis. Research in this area should consider the various subtypes of adenomyosis as well as the characteristics of bleeding.

This review of the existing literature suggests that future research in the field should take into consideration the challenges that have hindered research so far:(1)The still-unanswered question of defining normality and deviations therefrom.(2)The need for a classification of abnormal bleeding that has relevance to its etiology.(3)The need to explore the link between the presence of abnormal bleeding and its possible etiology and pathophysiology.

## 5. Conclusions

In conclusion, the relation between adenomyosis or any of its subtypes and AUB remains unproven or yet to be consolidated. The prevalence of the condition suggests that furthering our understanding of its clinical impact is an important area for research.

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
