# Peer review of "Adenomyosis and Abnormal Uterine Bleeding: Review of the Evidence"

_biomolecules, 2024, doi:10.3390/biom14060616_

Round 1

Reviewer 1 Report

Comments and Suggestions for Authors

Remarks:

- in the tables and in the Results section, the data is displayed in the same way (as text). In one of the two sections, I would expect a more transparent presentation of the results,

- in the Discussion section, the authors refer to the findings of other researchers without citing references (lines 428 - 448),

- typing errors (lines 319, 333, 486)

Author Response

Reviewer 1

Remarks:

- in the tables and in the Results section, the data is displayed in the same way (as text). In one of the two sections, I would expect a more transparent presentation of the results,

Reply: the manner of presentation of the results is constrained by the nature of the information available. We have taken the comment on board and revised the tables in line with recommendation and included a 4th column with additional information.

- in the Discussion section, the authors refer to the findings of other researchers without citing references (lines 428 - 448),

Reply:We apologise for the omission. This has now been corrected by inserting the relevant references

- typing errors (lines 319, 333, 486)

Reply: Our apologies. These errors have been corrected

Reviewer 2 Report

Comments and Suggestions for Authors

First, the method of collecting citations does not match the purpose of this review. Moreover, the purpose and conclusion of this review do not match either.

If the purpose of this review is " to examine published literature concerned with the relation between adenomyosis and AUB and to assess whether the introduction of the FIGO classification has enabled better understanding of this relation more than a decade since it was first proposed.," then a literature search should be done on AUB and Adenomyoisis. Most of the references used in this manuscript do not show the relationship between AUB and adenomyosis. Also, if the purpose of this manuscript is " whether the introduction of the FIGO classification has enabled better understanding of this relation more than a decade since it was first proposed.," the authors should separate the articles before and after 2011 and describe how it has changed.

The conclusion then states, " in understanding the relationship between AUB and adenomyosis, the unresolved problem of defining normal, the need for a scientific approach to the classification of abnormal bleeding, and the challenge of improving the accuracy of the classification of adenomyosis remain." and is not consistent with the purpose of this review. In addition, the conclusion of the text states, "There are three difficulties in research dealing with AUB: 1) The as yet unanswered question of the definition of normal and deviations from it. 2) The need for a classification of abnormal hemorrhage in relation to its etiology. 3) The need to explore the relationship between the presence of abnormal bleeding and its possible etiology and pathophysiology.

Some of the information does not match with [Summary Conclusions] and [conclusion of the text]. They should be unified.

Furthermore, the results are a list of short summaries of each article, and the manuscript is easy to read for the reader, but as a review article, it should state what can be said from these references.

Author Response

Reviewer 2

First, the method of collecting citations does not match the purpose of this review. Moreover, the purpose and conclusion of this review do not match either.

If the purpose of this review is "to examine published literature concerned with the relation between adenomyosis and AUB and to assess whether the introduction of the FIGO classification has enabled better understanding of this relation more than a decade since it was first proposed" then a literature search should be done on AUB and Adenomyosis.

Response: We thank the Reviewer for pointing to the major issue in our review.

Our review shows that unfortunately, since 2011, there has been no advance of our understanding of the relation between Adenomyosis and AUB. We have made this point clear in the discussion. The literature search was undertaken using MeSH terms to enable retrieval of all relevant articles, taking into consideration the changes and sometimes the non-specificity that the authors used when addressing uterine bleeding and its abnormalities.

To clarify this point we have added a section that addresses how FIGO classification was used in the cited articles.

Most of the references used in this manuscript do not show the relationship between AUB and adenomyosis.

Response: We agree with the reviewer! In fact, this is a main conclusion of our review. Specifically, we are fully aware that most published work in the field has taken for granted that Adenomyosis causes AUB, rather than address the issue as a question: “Is adenomyosis linked to AUB? Thus, most articles addressed the question only tangentially. By including all articles that reference “AUB and adenomyosis” we believe to have made sure that the reader has a clear knowledge of the situation, as well as to ensure that no relevant information is missed.

Also, if the purpose of this manuscript is "whether the introduction of the FIGO classification has enabled better understanding of this relation more than a decade since it was first proposed” the authors should separate the articles before and after 2011 and describe how it has changed. 

Response: We thank the Reviewer for this excellent recommendation. We have made this clearer in the text and indicated it in the results section.

The conclusion then states, "in understanding the relationship between AUB and adenomyosis, the unresolved problem of defining normal, the need for a scientific approach to the classification of abnormal bleeding, and the challenge of improving the accuracy of the classification of adenomyosis remain." and is not consistent with the purpose of this review. In addition, the conclusion of the text states, "There are three difficulties in research dealing with AUB: 1) The as yet unanswered question of the definition of normal and deviations from it. 2) The need for a classification of abnormal hemorrhage in relation to its etiology. 3) The need to explore the relationship between the presence of abnormal bleeding and its possible etiology and pathophysiology.

Some of the information does not match with [Summary Conclusions] and [conclusion of the text]. They should be unified.

Response: We have modified the conclusion section to take into account the points made by the reviewer.

Furthermore, the results are a list of short summaries of each article, and the manuscript is easy to read for the reader, but as a review article, it should state what can be said from these references.

Response: We thank the Reviewer for the suggestion which we have incorporated in the modified tables.

Reviewer 3 Report

Comments and Suggestions for Authors

This article by Habiba et al. is well done. It is a very comprehensive contribution to determine what we have and what is needed to understand adenomyosis and the clinical consequences. There is a need for more basic research and no additional meta-analyis. Patients have to be treated anyway.

Author Response

Reviewer 3

This article by Habiba et al. is well done. It is a very comprehensive contribution to determine what we have and what is needed to understand adenomyosis and the clinical consequences. There is a need for more basic research and no additional meta-analysis. Patients have to be treated anyway.

Response: thank you for the constructive comment and for fully grasping the conclusions and lessons from this review.

Round 2

Reviewer 2 Report

Comments and Suggestions for Authors

The relationship between the objectives, results, and conclusions of this Review is becoming appropriate and the manuscript is ready for publication. However, as we pointed out in the first issue, most of the references are listed in chronological order. They should be categorized and summarized according to the content of each study. The authors should state what can be said from these summarized references. In particular, the results in Table 1 need to be categorized according to the content of the paper. If this point is corrected, the manuscript may be accepted for publication.

Author Response

The relationship between the objectives, results, and conclusions of this Review is becoming appropriate and the manuscript is ready for publication.

Thank you. Much appreciated.

However, as we pointed out in the first issue, most of the references are listed in chronological order. They should be categorized and summarized according to the content of each study. The authors should state what can be said from these summarized references. In particular, the results in Table 1 need to be categorized according to the content of the paper. If this point is corrected, the manuscript may be accepted for publication.

Thank you for the helpful suggestion, we have taken this on board and revised table 1 into three broad categories. As you would appreciate, there is inevitable overlap. Nevertheless, we have revised and added a conclusion against each study within the text and for each of the broad groups.